# *Stenotrophomonas maltophilia* Infections: A Systematic Review and Meta-Analysis of Comparative Efficacy of Available Treatments, with Critical Assessment of Novel Therapeutic Options

**DOI:** 10.3390/antibiotics12050910

**Published:** 2023-05-15

**Authors:** Alberto Enrico Maraolo, Federica Licciardi, Ivan Gentile, Annalisa Saracino, Alessandra Belati, Davide Fiore Bavaro

**Affiliations:** 1First Division of Infectious Diseases, Cotugno Hospital, AORN Dei Colli, 80131 Naples, Italy; 2Department of Clinical Medicine and Surgery, Section of Infectious Diseases, University of Naples Federico II, 80131 Naples, Italy; federica.licciardi9@gmail.com (F.L.); ivan.gentile@unina.it (I.G.); 3Department of Biomedical Sciences and Human Oncology, Clinic of Infectious Diseases, University of Bari, 70124 Bari, Italy; annalisa.saracino@uniba.it (A.S.); alessandra.belati@hotmail.it (A.B.); davidebavaro@gmail.com (D.F.B.)

**Keywords:** trimethoprim/sulfamethoxazole, fluoroquinolones, minocycline, cefiderocol, antibiotic resistance, evidence synthesis

## Abstract

*Stenotrophomonas maltophilia* (SM) represents a challenging pathogen due to its resistance profile. A systematic review of the available evidence was conducted to evaluate the best treatment of SM infections to date, focusing on trimethoprim/sulfamethoxazole (TMP/SMX), fluoroquinolones (FQs), and tetracycline derivatives (TDs). Materials: PubMed/MEDLINE and Embase were searched from inception to 30 November 2022. The primary outcome was all-cause mortality. Secondary outcomes included clinical failure, adverse events, and length of stay. A random effects meta-analysis was performed. This study was registered with PROSPERO (CRD42022321893). Results: Twenty-four studies, all retrospective, were included. A significant difference in terms of overall mortality was observed when comparing as a monotherapy TMP/SMX versus FQs (odds ratio (OR) 1.46, 95% confidence interval (CI) 1.15–1.86, I^2^ = 33%; 11 studies, 2407 patients). The prediction interval (PI) did not touch the no effect line (1.06–1.93), but the results were not robust for the unmeasured confounding (E-value for point estimate of 1.71). When comparing TMP/SMX with TDs, the former showed an association with higher mortality but not significant and with a wide PI (OR 1.95, 95% CI 0.79–4.82, PI 0.01–685.99, I^2^ = 0%; 3 studies, 346 patients). Monotherapies in general exerted a protective effect against death opposed to the combination regimens but were not significant (OR 0.71, 95% CI 0.41–1.22, PI 0.16–3.08, I^2^ = 0%; 4 studies, 438 patients). Conclusions: Against SM infections, FQs and, possibly, TDs seem to be reasonable alternative choices to TMP/SMX. Data from clinical trials are urgently needed to better inform therapeutic choices in this setting by also taking into account newer agents.

## 1. Introduction

*Stenotrophomonas maltophilia* (SM) is an environmental Gram-negative aerobic bacterium that, acting mainly as an opportunistic pathogen, can cause a variety of clinical infections in various immunocompromised hosts, including subjects with genetic defects, such as cystic fibrosis, as well as individuals with weakened immune systems owing to cancer or other conditions leading to an immunosuppressed status [1].

Though with an intrinsic low virulence, the array of SM-associated human infections is vast: primarily respiratory tract infections and bloodstream infections (BSIs) but also endocarditis and infective processes affecting the eye, liver, nervous system, urinary tract, bone, soft tissue, and gastrointestinal tract, among others [2].

Known risk factors for SM infections are immunosuppression (especially linked with malignancies), chronic respiratory disease, indwelling devices, prolonged antibiotic use (mostly carbapenems), and long-term hospital stay or admission to intensive care units (ICUs) [3].

For sure, SM infections pose unique challenges to clinicians, since the pathogen harbors several intrinsic resistance mechanisms for frequently used antimicrobials and can further acquire resistance-encoding genes [4], therefore leaving a limited set of therapeutic options [5].

Trimethoprim/sulfamethoxazole (TMP/SMX) has been traditionally considered the cornerstone of the management of SM invasive infections, as it has historically showed the greatest in vitro potency against clinical isolates and been associated with better outcomes compared with other agents [6], although against a backdrop of considerable attributable mortality even when an appropriate initial antibiotic treatment has been implemented [7].

The lingering issue in defining the optimal treatment of SM infections has always been represented by the low quality of the underlying evidence, owing to the lack of randomized clinical trials (RCTs) and to other factors such as some shortcoming of the currently methods employed for antimicrobial susceptibility testing, the lack of clinical breakpoints, and the not constantly consistent correlation between in vitro data and clinical outcomes [8].

Notwithstanding these relevant limitations, an authoritative guidance document was released by the Infectious Diseases Society of America (IDSA) in December 2021 [9]. In summary, the IDSA paper suggests the use of the following antibiotics against SM infections: TMP/SMX, tetracycline derivatives (TDs) such as minocycline and tigecycline, fluoroquinolones (FQs), cefiderocol, and the association of ceftazidime/avibactam with aztreonam [9]. According to the IDSA, the preferred antibiotics are TMP/SMX and minocycline, to be considered the first-line options for severe infections, a scenario in which a combination therapy may be warranted [9]. Nevertheless, these well-known suggestions endorsed by the IDSA panel do not ensue from a systematic review of the literature. In light of the paucity of evidence syntheses on the topic of SM treatment, with only one article of this kind published before 2022 (dating back to 2019), contrasting FQs and TMP/SMX (no differences in clinical outcomes) [10], the purpose of this systematic review with a meta-analysis is to carry out a comprehensive comparison of the main options available against SM with regard to clinically relevant endpoints and, first and foremost, mortality. The quantitative analysis is accompanied by a narrative review aimed at describing the role of newer agents already contemplated by guidelines but not assessed in comparative studies.

## 2. Materials and Methods

### 2.1. Study Design

This study was conducted following the Preferred Reporting Items for Systematic Review and Meta-Analysis (PRISMA) guidelines on reporting systematic reviews and meta-analyses [11], specifically resorting to the latest version of PRISMA [12]. The corresponding checklist is provided in Appendix A (available in the Appendix A).

The research question was developed according to the PICO framework: population, intervention, comparison, and outcome [13]. The population was represented by patients with SM infections. Intervention was mainly constituted by TMP/SMX; if the studies did not assess the TMP/SMX, FQs were classified as the intervention. Monotherapy was considered among the interventions as well. Any drug different from TMP/SMX (or from FQs when the former was not involved) served as a comparison, with a special focus on TDs and cefiderocol; even combination therapy was considered when a treatment was administered as only one drug.

For the sake of simplicity, treatments diverse from TMP/SMX, FQs, TDs, and cefiderocol were grouped together as “other”.

The primary outcome was represented by all-cause mortality; the secondary outcomes were clinical failure, recurrence, length of stay (LOS), and safety.

The study protocol was submitted and registered with the PROSPERO International Prospective Register of Systematic Reviews (study ID: CRD42022321893) before the start of the literature search.

### 2.2. Eligibility Criteria

There were no limitations concerning language or geographical origin of the data.

Clinical studies reporting outcomes of interest related to patients with a SM infection were deemed eligible for inclusion, regardless of design (either interventional or observational); target age group (either adult or pediatric); setting (ICUs versus other wards); underlying comorbidities (e.g., malignancies); or type of infections (BSI, pneumonia, or other), as long as the following condition was met: the possibility of extracting data for at least two groups, each one having no less than 10 subjects, receiving different treatments (e.g., TMP/SMX versus FQs).

Preprints, abstracts, conference proceedings, commentaries, editorials, and review articles were excluded.

### 2.3. Data Source and Search Strategy

The following databases were searched from inception to 15 March 2022 and rerun up to 30 November 2022: PubMed/MEDLINE and Embase. Moreover, a hand search of the reference lists of all relevant reviews and original articles was performed. The detailed search strategy is described in Appendix A (available in the Appendix A). Appropriate keywords were combined, such as the name of the bacterium and the main antibiotics being compared.

If the retrieved articles did not include enough information about the outcomes under investigation, additional data from the corresponding authors were requested through e-mail.

### 2.4. Data Extraction

Two investigators (FL and AB) worked independently by screening each record for eligibility and inclusion using an electronic spreadsheet (Microsoft Corp., Redmond, WA, USA). The full texts of the articles judged eligible for inclusion were carefully assessed to establish the final list of works in the quantitative analysis and from which to extract relevant information. Any discrepancy from the two researchers was solved by consensus among the entire study group.

The subsequent data were abstracted: authors; country; publication year; type of study; timespan in which the study was run; number of patients; baseline features of the population under investigation (such as mean/median age, proportion of male/female subjects, and main comorbidities); follow-up timing; type of infection; type and schedule of antimicrobials; proportion of polymicrobial infections; outcome measures; and prognostic covariates (in the case of the multivariable analysis generating adjusted effect measures).

### 2.5. Outcomes Assessed

The primary outcome was all-cause mortality. The preferred timing was 28 or 30 days, but, in addition to in-hospital mortality, both shorter and longer follow-ups were taken into account in order not to exclude a priori studies in specific settings—for instance, in pediatric patients.

As stated above, the secondary outcomes were clinical failure, recurrence, LOS, and safety. The definitions used in the primary studies for these endpoints were adopted and explicitly stated.

As far as clinical failure was concerned, high heterogeneity in its definition was anticipated. In cases where clinical success was reported, the failure rate was computed accordingly.

About LOS, to ensure consistency, the reference parameter was hospital-related stay, and only alternatively other measures were taken into account, such as ICU-related and infection-related stays.

Similarly, regarding safety, to guarantee coherence among data, the reference parameter was represented by serious adverse events (AEs) or drug-related discontinuation and only alternatively by other reported endpoints.

### 2.6. Quality Assessment

Two reviewers (AEM and DFB) independently evaluated the study quality of the selected papers using of prespecified tools, and any disagreement was solved by general consensus. According to the study protocol, the Cochrane risk of bias tool for clinical trials in its updated version (RoB2) was meant to be used for randomized trials [14]. Observational studies were appraised through an adapted version of the Newcastle-Ottawa Scale (NOS) [15]. In this scale, observational studies were scored across three domains: selection (four questions), comparability (two questions), and ascertainment of the outcome of interest (three questions). Studies with fewer than 5 stars were considered of low quality (or a high risk of bias), 5 to 7 stars of moderate quality (and the same applies to the risk of bias), and more than 7 stars of high quality (low risk of bias). In addition to the score, a downgrading was performed if the study was not comparative.

### 2.7. Statistical Analyses

For each study included in the quantitative analysis, the effect size was represented by an odds ratio (OR) for binary outcomes and mean difference (MD) for continuous outcomes calculated with their 95% confidence intervals (CIs) and their 95% prediction intervals (PIs), an index that reflects the variations in treatment effects in different settings, including what effect is to be expected in future studies [16]. Meta-analyses were conducted through a random effects model (DerSimonian and Laird) in order to generate a pooled effect size due to the anticipated elevated heterogeneity among the studies in terms of the study design and comparators [17]. The weight in each study was calculated representing the inverse of the variance (the square of the standard error) of the study’s summary statistics. If not reported in the included records, the means and standard deviations for continuous outcomes were computed from the sample size, median, IQR, and minimum and maximum values, as described by Wan and colleagues [18]. The heterogeneity between studies was assessed by the I^2^ index, ensuing values of 25%, 50%, and 75% indicating low, moderate, and high heterogeneity, respectively [19]. Egger’s linear regression was employed to quantitatively evaluate the publication bias, which was also qualitatively gauged through funnel plots.

When adjusted data of the primary outcome were available, they were analyzed using the inverse variance method. The adjusted OR (aOR) was the effect size of choice for the pooling of adjusted data.

The anticipated subgroup analyses (when feasible) concerned variables such as study design (e.g., RCTs versus observational studies and comparative versus non-comparative) and place, different timing of mortality, adult versus pediatric population, hematological patients versus other type of subjects, type of infection (e.g., bloodstream infection or pneumonia), and the type of drug (e.g., trimethoprim-sulfamethoxazole versus each kind of fluoroquinolone).

For the sensitivity analysis, how each individual study impacted on the overall estimate was evaluated by a leave-one-out meta-analysis, thus generating influential plots. Furthermore, using quantitative bias analyses to assess the robustness of the results, the E-value was calculated, defined as the minimum strength of association that an unmeasured confounder would need to have with both the treatment and the outcome to completely explain away a specific association [20].

Eventually, meta-regression analyses were planned to investigate the potential study-level sources of heterogeneity; continuous moderators of interests were age and the proportion of polymicrobial infections, as long as at least ten studies for the covariate were retrieved.

Statistical significance was set at *p* < 0.05. All statistical analyses were performed with R software version 4.1.0 (R Foundation for Statistical Computing, Vienna, Austria) using the *meta* and *metafor* packages.

### 2.8. Narrative Synthesis

In addition to a quantitative analysis of the available options for SM infections based on their comparison, a qualitative description of the place in therapy of the newest agents was made.

### 2.9. Ethics

This work relies on previously approved and conducted studies, thus being exempt from ethics approval.

## 3. Results

### 3.1. Literature Search

In Figure 1 the study selection process is summarized as a flow diagram. A total of 2427 records, retrieved from two databases, were screened. After deduplication, 844 records were removed. From hand-searching four additional records identified in the meta-analysis by Ko and colleagues [10], who contacted the authors of some non-comparative studies to obtain unpublished data about mortality related to antibiotics, were added. Eventually, 24 studies were included in the quantitative analysis [21,22,23,24,25,26,27,28,29,30,31,32,33,34,35,36,37,38,39,40,41,42,43,44], out of 96 that were evaluated as full texts: the reasons for exclusion are described in the flow diagram.

### 3.2. Study Description

The baseline characteristics of the 24 studies included in the systematic review are presented in Table 1. All of them were observational in nature and retrospective. Less than half (10 out of 24) were comparative (between two or more options against SM) in scope. One-third of the studies were conducted in the United States (8/24). The largest study was the one by Sarzynski and colleagues [44] that provided a total of 1581 patients. On the other hand, the study that contributed the least (25 subjects) was the one by Ebara and collaborators [30]. In Table 1, further information is provided, such as the inclusion/exclusion criteria for each study, populations’ features, type of infections, administered drugs, dosages, administration as a monotherapy or in a combination, outcomes’ definitions, and the corresponding figures expressed as percentages.

### 3.3. Outcomes: Overview

Overall, the following comparisons were feasible: (i) about mortality, TMP/SMX versus FQs, TMP/SMX versus TDs, TMP/SMX versus others, FQs versus TDs, and monotherapy versus combination therapy; (ii) about clinical failure, TMP/SMX versus FQs, TMP/SMX versus TDs, TMP/SMX versus others (a miscellanea of less frequently used options), and FQs versus TDs; (iii) with regard to safety, TMP/SMX versus FQs; and (IV) with regard to LOS, TMP/SMX versus FQs and TMP/SMX versus TDs.

Moreover, as far as mortality is concerned, the pooling of adjusted effect sizes regarding the comparison between TMP/SMX and FQs was possible.

Recurrence of infection was addressed by only two studies [24,36]; considering that they involved different comparisons, a meta-analysis was not performed.

A complete overview of the results is provided in Table 2, where each effect size is accompanied by PIs, E-values, and comments when needed. Hereafter, a synthetic account of the main results stratified by the outcomes follows.

### 3.4. Mortality

The majority of studies focused on the contrast between TMP/SMX and FQs. To this purpose, different analyses were run. Of note, since only one study assessed ciprofloxacin alone as the FQ [37], other ones relying on levofloxacin or a mixed use of FQs, no subgroup analyses according to the different types of FQs were undertaken.

First, a comparison when drugs were administered as a monotherapy, which results are depicted in Figure 2: against a backdrop of 390 deaths out of 2407 patients (16%), the risk of mortality was higher with TMP/SMX in a statistically significant way, with an OR 1.46 (95% CI 1.15–1.86), in a context of modest heterogeneity (I^2^ = 33%) and with a PI neither including a null effect nor one opposite (1.10–1.93). No interaction was identified between the subtotal estimates for the three identified subgroups stratified by the timing of mortality assessment, thus confirming the null hypothesis that homogeneity existed between the different subgroup estimates of the population parameters.

This result, relying on a meta-analysis of crude, unadjusted data, was substantially confirmed when pooling the adjusted effect sizes. Indeed, in Figure 3, a related meta-analysis is illustrated; the number of underlying studies was lower (3 versus 11 in the main analysis), but the number of investigated patients was not so distant (1912 versus 2407). In essence, the FQ use was protective towards mortality: OR 0.73 (95% CI 0.56–0.95), the heterogeneity being negligible (I^2^ = 0%) but with a wide PI (0.13–4.10).

In the subgroup of the BSI, TMP/SMX monotherapy was again inferior compared with FQ monotherapy but not in a statistically significant manner (Table 2).

When also including studies in which either TMP/SMX or FQs could be used in the context of combination therapy, the worse outcome associated with TMP/SMX-based regimens was further corroborated, as shown in Figure 4: OR 1.58 (95% CI 1.10–2.27), I^2^ = 43%, including more patients (2806, with an overall higher death rate equal to 20%); nevertheless, the PI ranged from 0.58 to 4.35, including an opposite effect as well. No interaction was demonstrated between the subtotal estimates for the four identified subgroups based on mortality timing. In the subgroup of the BSI, TMP/SMX-based regimens were linked with a more than two-fold higher risk of a fatal outcome compared with the FQ-based regimens (Table 2).

TMP/SMX was worse than the other two kinds of comparators: TDs and other drugs, although not significant in either case (Table 2).

Based on the data from only 174 subjects, FQ use was protective towards mortality, even against TDs, but the results were not significant, and the PI was extremely wide (Table 2).

Eventually, when the monotherapy strategy was compared with a combination approach, whichever were the anti-SM agents, the former was associated with a better outcome compared to the latter (Figure 5): OR 0.71 (95% CI 0.56–0.95) against a backdrop of no heterogeneity (I^2^ = 0); the PI also, in this case, contained the opposite effect (0.16–3.08), but no subgroup difference was highlighted among the adult and pediatric patients.

### 3.5. Clinical Failure

A minority of the studies addressed clinical failure as the outcome. In all available comparisons, no significant differences from a statistical viewpoint were detected, and the small number of included patients favored very ample or even incalculable PIs (Table 2).

### 3.6. Safety

About safety, only one comparison was feasible: TMP/SMX versus FQs, always in a monotherapy. The risk of an adverse event was nearly double with the former as opposed to the latter: OR 1.89 (95% CI 0.26–13.60), I^2^ = 81%; the PI was extremely wide (Table 2).

### 3.7. Length of Stay

Persistence with regard to a LOS, TMP/SMX was associated with a longer duration of hospital stay compared with FQs and, also, with TDs, although with large and even incalculable PIs, respectively (Table 2).

### 3.8. Sources of Heterogeneity and Sensitivity Analyses

To investigate sources of heterogenetic results, subgroup analyses were carried out. Some of them were already presented in the main analyses—for instance, mortality according to the different reporting times. The subgroups of patients affected by BSI were already reported as well.

All studies were observational in nature, so no subgroup analysis was performed regarding the study design. Nevertheless, a comparison between TMP/SMX and FQs according to the comparative or non-comparative nature of the study was feasible, and their results are displayed in Appendix A (both available in the Appendix A), the former regarding only monotherapy studies and the latter also concerning studies allowing associations based either on TMP/SMX or on FQs. In both cases, the test for subgroup differences indicated that there was no statistically significant subgroup effect (*p* = 0.60 and *p* = 0.4 analyses not presented), suggesting that the type of study does not modify the effect of one option in comparison to another one. Moreover, in both instances, the OR for mortality associated with TMP/SMX was higher in the subgroup of non-comparative studies but was statistically significant solely in the subgroup including comparative studies.

The data did not allow being split according to variables such as adult versus pediatric patients, immunosuppressed/hematological/ICU subjects, or eventually, specific subtypes of drugs within the same class (FQs and TDs).

Additionally, the planned meta-regression analyses involving continuous moderators such as age and the proportion of polymicrobial infections were not feasible; the number of studies reporting proper data was too low, considering that, in many cases, especially when non-comparative studies were addressed, the available information relied on an entire population from which a fraction of subjects was excluded for the present analysis (e.g., subjects without active therapy); therefore, granular data centered exclusively on the included patients were missing.

A sensitivity analysis through the leave-one-out method was performed, as far as the primary outcome was concerned, only for the comparison between TMP/SMX and FQs, the one based on more studies, and the related results are shown in Appendix A (both available in the Appendix A). In the first case (only monotherapy studies), omitting the work by Sarzynski and colleagues [44], the largest as the sample size, did not shift the direction of the effect but made the results not significant: OR 1.65 (95% CI 0.98–2.78), with nearly overlapping heterogeneity (I^2^ = 39%). Instead, omitting the study by Wang CH and collaborators [28], the one describing the worst outcome from TMP/SMX, shrank the heterogeneity to zero without impacting the results. In the second case, neither the effect size nor the heterogeneity were notably affected by omitting a particular study.

Further sensitivity analyses revolved around the study by Sarzynski and colleagues [44] presented in Appendix A (available in the Appendix A). Specifically, analyses regarding the primary outcome were rerun by excluding other studies conducted in the United States [25,33,35,39], since Sarzynski’s work drew data from national databases (the period ranging from 2005 to 2017) that potentially overlapped with the timespan in which previous studies were conducted in the same country [44]. As shown in Appendix A, the direction and magnitude of the effect size were not impacted when contrasting TMP/SMX and FQs either as a monotherapy or by not removing four studies. Moreover, for the exclusion, the study by Junco and colleagues [39] did not change the results of the pooled adjusted ORs. The last sensitivity analysis was based on the reconstruction of the 2 × 2 contingency table, according to a method described elsewhere starting from the OR and from the total number of patients in each arm, as well as from the total number of events [45], concerning the BSI subgroup in Sarzynski’s paper, that did not provide a raw number of dead subjects and survivors stratified by treatment in the population with BSIs [44]. At any rate, the addition of these imputed data did minimally modify the results of the comparison between TMP/SMX and FQs, either as a monotherapy or not, in the BSI subgroup; the magnitude of the effect was slightly reduced, but TMP/SMX remained associated with a more than two-fold risk of mortality as opposed to the FQs.

A quantitative bias analysis demonstrated that all results were not robust in the unmeasured confounding; the E-value, a measure assessing the plausibility that an association could be explained away by residual confounding, was a small entity for most of the comparisons, and in many cases, the value for the CIs was simply 1, so no confounding was needed to move the CIs to include 1 (Table 2).

### 3.9. Publication Bias and Quality Assessment

In the Appendix A, two contour-enhanced funnel plots are depicted (Appendix A) to detect small-study effects as a proxy of the publication bias, in order to show how asymmetry patterns relate to statistical significance, as far as the primary outcome was concerned in the contrast between TMP/SMX and FQs (the one based on more studies) exclusively as a monotherapy or not, respectively. The funnel plots were quite symmetrical; beyond an inspection, the absence of asymmetry in the funnel plot was confirmed quantitively by Egger’s regression test, which results were not statistically significant: *p* = 0.46 and *p* = 0.29, respectively.

The risk of bias was assessed by resorting to only one tool, the NOS, in light of the same study design across all the included studies. In Appendix A (available in the Appendix A), the results of this assessment are illustrated; in total, 14 studies were judged to be of low quality (high risk of bias), especially for their noncomparative natures, 7 of moderate quality, and just 1 of high quality, the work by Sarzynski and collaborators [44].

## 4. Discussion

To the best of our knowledge, the present study is the most comprehensive systematic review addressing the topic of the treatment of SM infections, including a quantitative assessment of as many comparisons between available therapeutic options as possible. A thorough narrative description of most of the studies included, accompanied by insightful comments, is provided elsewhere [8].

In the recent past, the main evidence synthesis published focused on the contrast between TMP/SMX and FQs [10]. The meta-analysis by Ko and co-workers, whose search covered up to March 2018, concluded that FQ use was associated with survival benefits compared to TMP/SMX: OR 0.62 (95% 0.39–0.99). That work only included 663 patients from 14 observational studies (7 retrospective cohorts and 7 case–control); in some cases, the denominator in either arm was inferior to 10, and the majority of the included studies were not comparative in nature, such that 71% of cases had pooled unpublished information on mortality according to different antibiotic regimens (sent privately by the authors of the original works). Our analysis, although always depending only on observational and retrospective data, yielded a result similar in magnitude in favor of FQs but corroborated by elements such as (opposed to the previous paper) a larger sample size, distinction between different scenarios (only a monotherapy or associations allowed, comparative versus non-comparative studies), a focus on BSIs, and the availability of adjusted effect sizes; as above mentioned, when pooling aORs, the benefits of FQs over TMP/SMX were confirmed, with statistically significant results.

Nevertheless, the interpretation of these results should take careful consideration regarding the role of the large study conducted by Sarzynski and colleagues [44]. By performing a sensitivity analysis with the leave-one-out method, the benefits lost their statistical significance, at least in the context of only monotherapy studies, although a trend in favor of FQs was still apparent. Indeed, their research group conducted the largest retrospective study currently published on SM infections, collecting data from 154 hospitals across the United States and including 1581 patients [44]. The study was conceived to compare levofloxacin and TMP/SMX as monotherapies: 823 patients were treated with the former and 758 patients with the latter, respectively. The overall mortality was 16.4%, significantly higher in patients with a low respiratory tract infection (19.5%) compared with the BSI (14.1%). The study employed a series of elegant statistical techniques to mitigate biases linked to its observational nature, resorting to overlap weighting as the propensity score method to adjust for confounding due to differences between comparator groups and resorting to adjusting for the time to culture as a continuous variable to account for a potential immortal time bias. Overall, the benefit of levofloxacin was sensible (risk reduction of nearly 25%) but not statistically significant: aOR 0.76, 95% CI 0.58–1.01. The benefit became apparent in the subgroup of patients affected by a low respiratory tract infection that was largely the most represented, including 90.9% of patients (1418/1561): aOR 0.73, 95% CI 0.54–0.98. In the BSI group, on the contrary, the FQ use was associated with a worse outcome, implying a notable uncertainty in the estimate: aOR 1.18, 95% CI 0.47–3.02.

This last finding was in contrast with the results of the present systematic review, in which BSI was the only homogenous subgroup that could be the object of specific investigation into the framework of the comparison between TMP/SMX and FQs; even considering the data by Sarzynski and colleagues [44], FQs were linked with increased survival.

The only other meta-analysis on different therapeutic approaches against SM infections was published very recently and focused on the role of combination therapy versus monotherapy [46]. Starting from different and less strict inclusion criteria and by using a different research string, the authors included four studies, of which only one was similar to the corresponding analysis in the present work [36], by pooling unadjusted effect sizes to produce an overall hazard ratio (HR) for mortality; combination therapy fared better in the BSI subgroup (HR 0.76, 95% CI 0.18–3.18, two studies) and worse among pneumonia patients (HR 1.42, 95% CI 1.04–1.94, two studies). In our analysis, comprising both adult and pediatric patients from the same limited number of studies (four), a trend emerged in favor of monotherapy, even if not significant.

Currently, the use of monotherapy versus combination therapy is still debated in cases of Gram-negative infections. For instance, a meta-analysis published in 2019 on Gram-negative infections treated with older drugs showed a superiority of combination therapy, especially in the case of BSI caused by carbapenemase-producing bacteria and Acinetobacter baumannii [47], probably because active beta-lactams for these pathogens were unavailable when the studies were performed.

Nevertheless, the subsequent meta-analysis and large cohort studies recently published on Pseudomonas aeruginosa and KPC-producing Klebsiella pneumoniae have shown that combination therapy did not confer a real benefit but was burdened by a higher number of adverse events [48,49].

At any rate, robust evidence on the use of combination therapy versus monotherapy in cases of SM infection is lacking; as a consequence, this meta-analysis, as the one already published, cannot provide solid suggestions regarding this point.

Eventually, in the present meta-analysis, other antibiotic regimens were evaluated as well, including TDs such minocycline and tigecycline.

Actually, sparse data were available even for other agents, such as ceftazidime, ticarcillin/clavulanate, and polymyxins from non-comparative studies [21,32,40,41]. Data were lumped together in order to carry out a comparison based on sufficient data with TMP/SMX and also including the aggregation of TDs and FQs in one study [39]. TMP/SMX use was linked with slightly worse outcomes, but the result was not significant and must be interpreted with caution in light of the nature of the data and of the aggregation of very different therapeutic choices.

When taking into account TDs, two types of contrasts were possible: against TMP/SMX and FQs. In both cases, only a few studies could be pooled, with limited numbers of patients (346 and 174, respectively). TMP/SMX once again fared worse than the comparator, being associated with a nearly two-fold risk of a fatal outcome, whereas the FQs seemed to exert a protective effect; in both scenarios, the results were not significant, and considering the high underlying heterogeneity conveyed by very large PIs, no firm conclusion could be made.

As hinted in the introduction, the IDSA guidelines recommend TMP/SMX as the drug of choice against SM, suggesting using it in a combination regimen in cases of moderate to severe infections and even sequentially after starting as a monotherapy [9]. All other options are considered the second line [9].

Taken at face value, the results of the present meta-analysis potentially pave the way for relevant changes to the current recommendations about the management of SM infections that are informed by preclinical data, expert opinion, and low-quality studies. Of course, this quantitative review of the literature is also influenced by the well-known limitations characterizing studies beyond SM. Therefore, the results need to be carefully interpreted within a comprehensive clinical and microbiological context.

Firstly, especially in some scenarios, the real meaning of SM identification from a culture remains elusive, whether a true pathogen or a bystander/colonizer:;this applies, for instance, when the sample is from the respiratory tract of subjects with structural lung disease (e.g., cystic fibrosis) or those with ventilator dependance; to some extent, it applies in cases of suspected BSI when central lines are involved as well [50]. Mostly in older studies, this might have inflated the benefit of drugs such as ceftazidime, the only agent with approved but outdated breakpoints by the Food and Drug Administration [51] and which successes were probably related to the treatments of colonization and not of true infections. For drugs such as ceftazidime, breakpoints likely do not accurately represent the impact of some resistance mechanisms in vivo; the correct interpretation of colistin susceptibility tests may be hindered by heteroresistance [52]; this is why the IDSA does not recommend these drugs [9] and the reason why the results of the present meta-analysis concerning these agents as an aggregated group should be considered with caution.

Secondly, SM is often identified in the context of polymicrobial infections [53], thus rendering it even more difficult to attribute a definite pathogenetic role to SM itself and to disentangle the therapeutic effectiveness of anti-SM agents when multiple drugs are used at once to target other organisms. It is worthwhile noting that one of the main exclusion criteria in Sarzynki’s study was the receipt of any antimicrobial with known in vitro activity against SM different from TMP/SMX and levofloxacin, increasing the robustness of its findings in favor of the FQ [44].

Thirdly, the issues concerning antimicrobial susceptibility testing (AST) cannot be overlooked, because data to support a relationship between susceptibility testing results and the clinical outcome with SM infections are lacking for many agents. In Europe, since 2012, the European Committee on Antimicrobial Susceptibility Testing (EUCAST) has released breakpoints only for TMP/SMX (resistance for values higher than 4 mg/L, expressed as the trimethoprim concentration) owing to the AST difficulties stemming from many factors potentially influencing the results, such as incubation temperature, culture medium, and technique [54]. In the United States, the Clinical and Laboratory Standards Institute (CLSI) provides breakpoints for TMP/SMX, levofloxacin, minocycline, ceftazidime, ticarcillin/clavulanate, chloramphenicol, and cefiderocol [8]. Very recently, a designated working group summoned by the CLSI to revise the clinical breakpoints decided not to lower the one for levofloxacin from 2 mg/L to 1 mg/L, according to the results of a neutropenic murine thigh infection model [55], but a note was added to use the drug in association, pending new insights from Sarzynki’s cohort, stratifying the outcome by using the pathogen minimum inhibitory concentration (MIC) [56].

Fourthly, the AST issues intermingle with real-word susceptibility data and with pharmacokinetic/pharmacodynamic (Pk/Pd) considerations. The SENTRY program quite recently reported the overall susceptibility of 6467 SM isolates from a worldwide collection (from 1997 to 2016) by using reference standard broth microdilution (BMD) [57]. The most active agent test against SM was minocycline (99.5% of strains susceptible according to the CLSI criteria), followed by TMP/SMX (96.2% of isolate susceptible, according to the EUCAST criteria, a proportion stable over two decades) and tigecycline and levofloxacin (81.5% of strains susceptible, CLSI criteria) [57]. The fact that most commercial testing systems yield interpretations with a high post-test probability of being accurate for TMP/SMX reinforces its role as the mainstay of anti-SM therapy, whereas errors are not infrequent in levofloxacin testing when commercial systems are used [58]. Another reason for concern about FQs stems from the several mechanisms of resistance displayed by SM: the chromosomally encoded qnr gene, mutations of the bacterial topoisomerase and gyrase genes, and the efflux pump SmeDEF that, in turn, protect both gyrase and topoisomerase IV from FQs [59]. Moreover, exposure to FQs can generate resistance not only to FQs themselves but also to other frontline anti-SM agents [60]. Nevertheless, in a clinical scenario such as pneumonia, the more favorable Pk/Pd properties of levofloxacin compared to TMP/SMX should be taken into account: a faster time-to-peak serum concentration, higher concentration in epithelial lining fluid, and bactericidal activity [3,61]. These may have been the driving factors of the better survival associated with FQs in Sarzynski’s study [44], but in the present meta-analysis, the signal of the potential superiority of FQs compared to TMP/SMX was interestingly apparent in the BSI subgroup as well. With regard to TDs, as mentioned before, tigecycline displays reliable in vitro activity, but its use is limited, since it showed an increasing trend in clinical failure, mortality, and adverse events in several real-life studies, probably due to its Pk/Pd properties [62]. On the other hand, minocycline might be an interesting alternative to TMP/SMX: beyond displaying the best activity against SM isolates [57], according to limited Pk/Pd data, a high-dose regimen (200 mg twice daily) provides the highest probability of target attainment across its MIC distribution [8]; lastly, minocycline shares with TMP/SMX the availability of commercial testing systems generating accurate results [58].

## 5. New Therapeutic Options

Despite their potential activity, the use of molecules such as TMP-SMX and FQs for the treatment of Gram-negative bacterial infections is generally burdened by more unfavorable outcomes compared to the use of new molecules due to the more frequent adverse events, high selective pressure, and often suboptimal Pk/Pd features, especially for BSIs. Against this backdrop, new molecules that can represent additional weapons against SM infections have not been tested yet in randomized clinical trials or in real-life studies.

Several new agents are in development to treat Gram-negative infections, but few options appear effective against SM.

Cefiderocol is a new siderophore cephalosporin recently approved for the treatment of infections due to aerobic Gram-negative organisms in adults with limited treatment options. Stracquadanio and colleagues conducted an in vitro study evaluating MICs of cefiderocol against SM on 127 isolates, finding that cefiderocol displayed the lowest MIC values with 100% efficacy on the tested strains compared to colistin, ceftazidime/avibactam, and ceftolozane/tazobactam, which showed less susceptibility [63]. Additionally, in animal models, cefiderocol appears to be promising [64], but real-life data are limited to the CREDIBLE-CR trial, which included only five cases of SM pneumonia in the cefiderocol group, with a survival rate of 80% (4/5) [65].

An alternative therapeutic option to cefiderocol could be represented by aztreonam combined with a beta-lactam/beta-lactamase inhibitor (avibactam, clavulanate, relebactam, or vaborbactam), with the theoretical aim of overcoming the intrinsic expression of L1 metallo- and L2 serine-beta-lactamases by SM. Indeed, Biagi and collaborators tested the in vitro activity of a combination of aztreonam with different beta-lactamases inhibitors on 47 isolates of SM resistant to levofloxacin and TMP/SMX. They found that avibactam restored aztreonam sensitivity in 98% of the isolates, while the combinations with other beta-lactamases inhibitors were less effective [66]. The data were also supported by a molecular analysis demonstrating a hyperexpression of L1 and L2 and the efflux pump (smeABC). However, in vivo data are limited to a few case series and case reports. A recent systematic review reported only 94 patients with MBL Gram-negative infections treated with ceftazidime/avibactam plus aztreonam, with a clinical resolution of 80% [67]. Based on this evidence, cefiderocol or aztreonam plus ceftazidime/avibactam has been suggested by a panel of IDSA experts as a potential monotherapy against SM infections [9].

Eventually, eravacycline and omadacycline are new tetracyclines that could represent a future potential treatment strategy. However, eravacycline has a Pk/Pd pattern similar to tigecycline, which may be disadvantageous in cases of BSIs; nevertheless, the data showed a significant antibacterial activity and a very wide spectrum of efficacy [68]. On the other hand, omadacycline also demonstrated potential in vitro activity, but the MICs appeared to be higher than the other TDs [69].

## 6. Limitations

The present meta-analysis has several limitations. Firstly, no RCT is available for SM infections. Only a few observational studies are available, with important imbalances in the sample sizes, and only a fraction of them compare outcomes associated with different therapeutic strategies through uni- or multivariable analyses. Therefore, this work inherited the limitations of the underlying evidence: residual confounding, immortal time bias, and confounding by indication. Since only observational studies were retrievable, no formal assessment of confidence in the body of evidence for each outcome was performed: it is implicit to categorize the ensuing evidence as having low/very low certainty. At any rate, many sensitivity analyses have been performed that have confirmed the signal of potential superiority of FQs compared to TMP/SMX. Secondly, the heterogeneity between studies was not negligible, and the studies were not perfectly comparable. Thirdly, further analyses conducted in our study, including clinical failure, adverse events, and LOS, were hampered by the paucity of data. Similarly, the aim of comparing different healthcare settings (ICUs, immunocompromised hosts, and so on) or different sites of infection (BSI, pneumonia, and so on) was also hindered by the high heterogeneity of the data and lack of sample size. This may limit the generalizability of the results; as a matter of fact, in the main analysis (TMP/SMX versus FQs), the overall rare deaths ranged from 16% to 20%, far below the mortality usually linked with SM (up to 37.5%) [7], suggesting that a nonnegligible fraction of patients was affected by not-severe infections. Fourthly, it was not possible to stratify the outcomes according to the MIC values for each antibiotic.

Of course, the lack of these data is an important limitation that should be deeply studied in the future, since several works have demonstrated that, excluding the number and type of antibiotics used, many other strategies should be implemented in cases of severe infections, targeting the host and their clinical conditions [70].

Indeed, in a recent study, the benefit of a multi-step bundles approach aimed at managing BSIs by Gram-negative bacteria as a “clinical syndrome” was demonstrated; mortality was reduced by improving the identification of deep sites of infections, the rate of early targeted antimicrobial therapy, and the rapid discontinuation of antibiotics in cases of uncomplicated BSIs [71]. These bundles should also be used in cases of SM infections; in fact, as occurred in the study conducted by Sarzynski and co-workers [44], patients in the levofloxacin group showed a reduced mortality probably because they were more likely to receive an early effective empirical therapy (10% versus 0.9%).

## 7. Conclusions

The present meta-analysis suggests that the use of FQs—in particular, levofloxacin—when the respiratory tract is involved might be a reasonable alternative to TMP/SMX for SM infections, even as a monotherapy. Importantly, this choice should be balanced with the risk of inaccurate testing results and emergent resistance by the selective pressure associated with FQ use. To some extent, even TDs—particularly high-dose minocycline—might serve as a first-line alternative to TMP/SMX. Rock-solid evidence recommending combination therapy is lacking. While the place of new molecules in therapy is better defined, it is urgent to set up well-conducted prospective observational studies and, most of all, RCTs to compare the currently available best treatment strategies against SM infections.

## Figures and Tables

**Figure 1 antibiotics-12-00910-f001:**
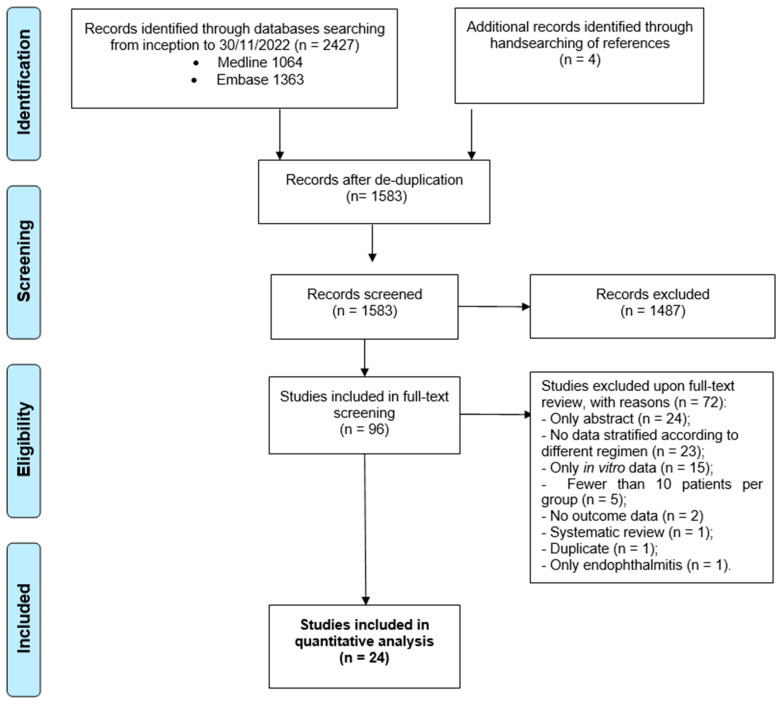
Results of the literature search, and a flow diagram for the selection of eligible studies.

**Figure 2 antibiotics-12-00910-f002:**
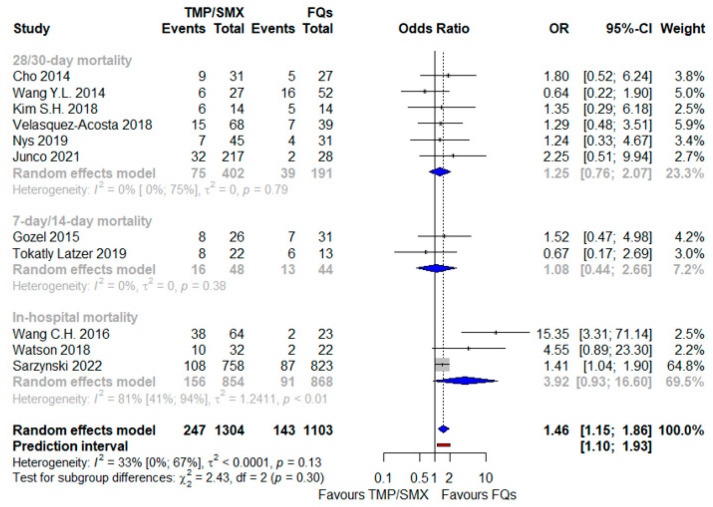
Forest plot showing the OR for mortality related to TMP/SMX use as opposed to FQ administration, both as a monotherapy [24,25,26,28,31,32,33,35,37,39,44]. Abbreviations: FQs: fluoroquinolones; OR: odds ratio; TMP/SMX: trimethoprim/sulfamethoxazole.

**Figure 3 antibiotics-12-00910-f003:**
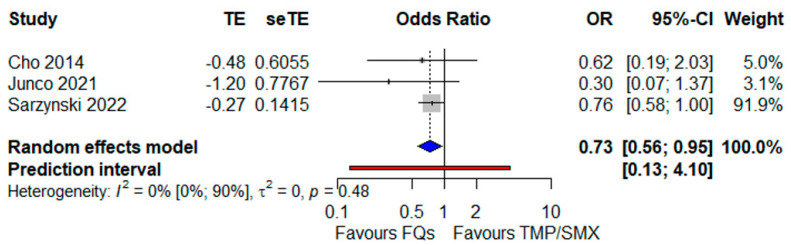
Forest plot showing the OR for mortality related to FQ use as opposed to TMP/SMX administration, both as a monotherapy [24,39,44]. Abbreviations: OR: odds ratio; FQs: fluoroquinolones; TMP/SMX: trimethoprim/sulfamethoxazole.

**Figure 4 antibiotics-12-00910-f004:**
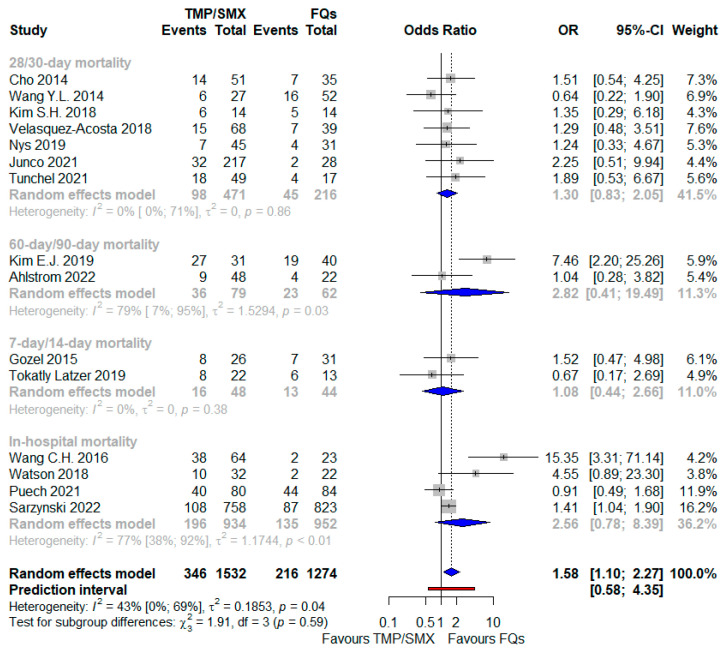
Forest plot showing the OR for mortality related to TMP/SMX use as opposed to FQ administration and also including studies taking into account a combination therapy [24,25,26,28,31,32,33,34,35,37,39,40,41,43,44]. Abbreviations: FQs: fluoroquinolones; OR: odds ratio; TMP/SMX: trimethoprim/sulfamethoxazole.

**Figure 5 antibiotics-12-00910-f005:**
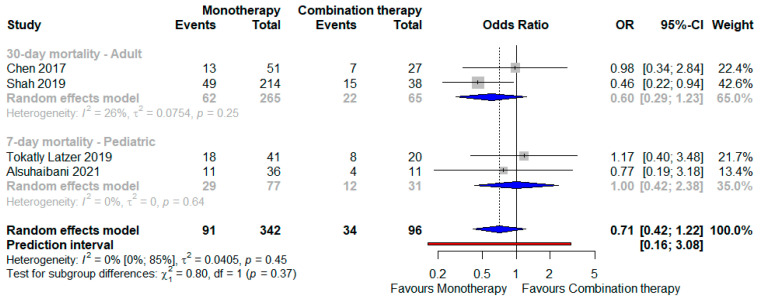
Forest plot showing the OR for mortality related to TMP/SMX use as opposed to FQ administration and also including studies taking into account a combination therapy [29,36,37,38]. Abbreviations: OR: odds ratio.

**Table 1 antibiotics-12-00910-t001:** General features of the included studies for quantitative synthesis.

Author, Year [Ref.]	Country, Design and No. Centers	Study Period	Main Inclusion and Exclusion Criteria	Type of Infection	Study Population	Group			Mortality (Definition)	Safety Assessment (Definition)	Length of Stay(Definition)	Multivariable Analysis on Mortality (Type and Variables)	Comments
						Reference Regimen (Number of Patients, Daily Dose, Treatment Duration, Combination Therapy [%])	Comparator (Number of Patients, Daily Dose, Treatment Duration, Combination Therapy [%, Drug])	Other Comparators (Number of Patients, Daily Dose, Treatment Duration, Combination Therapy [%, Drug]) If Any					
Garcia Paez et al., 2008 [21]	Brazil—Retrospective -single center	From July 1999 to July 2005	Inclusion: only adult patients. Exclusion: colonization and not infection by SM, medical record unavailable	BSI: 87% Pneumonia: 13%	Adults Male: 70% Mean age 48.9 years Malignancy: 45%	Regimen: TMP/SMX (Dosage not specified) Duration undefined	Regimen: others (Dosage not specified) Duration undefined	N/A	TMP/SMX: 26% Others: 46% (14-day)	Not addressed	Not addressed	Not addressed	Absence of therapy excluded from comparator; not clear if TMP/SMX used as monotherapy or not Polymicrobial infection: 30% Non-comparative study
Czosnowski et al., 2011 [22]	United States, Retrospective- Single center	January 1997–December 2007	Inclusion: only adult, ICU. Exclusion: incomplete medical record data.	VAP	Adults Male: 76% Mean age 40 years Traumatic brain injury: 56%	Regimen: TMP/SMX (11.2 ± 3.8 mg/kg/day) Duration undefined Combination allowed	Regimen: Others (Dosage not specified) Duration undefined Combination allowed	N/A	N/A	Not addressed	Not addressed	Not addressed	Polymicrobial infection: 66% No data on mortality, only on clinical failure Treatment failure defined as either clinical failure, or microbiologic plus clinical failure: TMP/SMX: 14% Others: 8% Overall treatment duration was 11.4 days (mean) Non-comparative study
Tekce et al., 2012 [23]	Turkey—Retrospective cohort—Single center	From January 2008 to December 2010	Inclusion: Patients who had received more than 3 days of TMP/SMX or tigecycline for nosocomial SM infection	Pneumonia: 51% SSI: 29%	Adults. Male: 53% Mean age 65.4 years Malignancy: 29% ICU stay: 87%	Regimen: TMP/SMX (Dosage not specified) Duration undefined 0% combination	Regimen: Tigecycline (Dosage not specified) Duration undefined 0% combination	N/A	TMP/SMX: 31% Tigecycline: 21% (30-day)	Not addressed	Not addressed	Not reported	Polymicrobial infection: 29 (64.4%) patients Comparative study
Cho et al., 2014 [24]	South Korea—Retrospective cohort—Single center	From 2000 to 2012	Inclusion: only adult patients. Exclusion: combination therapy between TMP/SMX and levofloxacin, death within the first 2 days after the start of the therapy	BSI	Adults Median age 58 years (IQR 45–67) Malignancy: 52% MV: 16%	Regimen: TMP/SMX 51-patients (15–20 mg/kg of body weight/day TMP) Duration undefined 18% combination	Regimen: Levofloxacin—35 patients (750 mg/day) Duration undefined 9% combination	N/A	TMP/SMX: 27% Levofloxacin: 20% (30-day)	TMP/SMX: 24% Levofloxacin: 0% (adverse events)	TMP/SMX: median 25 days (IQR 12–51) FQs: median 27 days (IQR 15–52) (hospital stay)	Levofloxacin use versus TMP/SMX: aOR 0.62 (95% CI 0.19–2.04) Adjusted was made for septic shock and pneumonia	Data could be stratified according to monotherapy and combination therapy. Polymicrobial infection: 20% Recurrence (30-day): TMP/SMX = 12% Levofloxacin = 6% Comparative study
Wang YL et al., 2014 [25]	United States—Retrospective cohort—Single center	From January 2008 to December 2011	Adult patients with nosocomial SM infection received monotherapy with TMP/SMX or an FQ for at least 48 h.	Pulmonary infection: 56% SSTI: 19% UTI: 9% IAI: 9% Secondary BSI: 6%	Adults Male: 61% Mean age: 73 years Solid organ malignancy: 39% MV: 30%	Regimen: TMP/SMX 35-patients (Dosage not specified) Median duration 8 days (IQR 2–28) 0% combination	Regimen: FQs—63 patients Levofloxacin = 76% Ciprofloxacin = 24% (Dosage not specified) Median duration 9 days (IQR 2–8) 0% combination	N/A	TMP/SMX: 22% FQs: 31% (30-day)	Not addressed	TMP/SMX: median 16 days (IQR 8–42) FQs: median 25 days (IQR 15–37) (hospital stay)	Not reported	Polymicrobial infection: 77% ICU admission at time of culture: 24% Comparative study
Gokhan Gozel et al., 2015 [26]	Turkey— Retrospective, Single center	From January 2006 to December 2013	Inclusion: only adult patients. Exclusion: polymicrobial infection	BSI: 49% Pneumonia 51%	Adults Male: 66% Median age 68 years (IQR 20–87) Malignancy: 24%	Regimen: TMP/SMX—26 patients (Dosage not specified) Duration undefined 0% combination	Regimen: Levofloxacin-31 patients (Dosage not specified) Duration undefined 0% combination	N/A	TMP/SMX: 31% Levofloxacin: 23% (14-day)	Not addressed	Not addressed	Not addressed	Unpublished data of the original article were retrieved from the paper of Ko et al. [10] Polymicrobial infection: 11% Non-comparative study
Hand et al., 2016 [27]	United States—Retrospective—Single center	From January 2006 to December 2012	Inclusion: adult and pediatric patients with; one positive culture for SM. Exclusion: combination therapy, concomitant antibiotics with anti-SM activity other than the ones studied.	Mixed	Male: 47% Mean age: 52 years (calculated combining two means) MV: 51%	Regimen: TMP/SMX—22 patients (average daily doses of 200 mg/day SMX and 8.5 mg/kg/day TMP) Median duration 7 days (IQR 3–15) 0% combination	Regimen: Minocycline-23 patients (200 mg daily) Median duration 14 days (IQR 4–12) 0% combination	N/A	TMP/SMX: 9% Minocycline: 9% (30-day)	Not addressed	TMP/SMX: median 54 days (IQR 4–265) Minocycline: median 41 days (IQR 6–136) (hospital stay)	Not addressed	Polymicrobial infection: 73% Treatment failure (isolation of SM on follow-up culture from the same site as the initial infection within 30 days of the initial culture or in-hospital death within 30 days of the initial positive culture or receipt of an alternative or additional antibiotic possessing in vitro activity against SM during any point of initial therapy): TMP/SMX = 39% Minocycline = 48% Comparative study
Wang CH et al., 2016 [28]	Taiwan—Retrospective—Single center	From January 2004 to December 2013	Inclusion: All patients with monomicrobial SM BSI. Exclusion: patients who had polymicrobial BSI or who were aged <18 years.	BSI	Adults Male: 73% Mean age: 68.3 years Malignancy: 38% MV: 64%	Regimen: TMP/SMX—64 patients (Dosage not specified) Duration undefined 0% combination	Regimen: FQs—23 patients (Dosage not specified) Duration undefined 0% combination	N/A	TMP/SMX: 59% FQs: 9% (in-hospital)	Not addressed	Not addressed	Not addressed	Unpublished data of the original article were retrieved from the paper of Ko et al. [10] Non-comparative study
Chen et al., 2017 [29]	China—Retrospective cohort—Single center	From January 2009 to March 2015	Inclusion: only adult patients. Exclusion: patients without adequate medical records or any clinical manifestation.	BSI	Adults Male: 64% Solid tumors: 26% ICU: 26%	Monotherapy—51 patients (Dosage not specified) Duration undefined	Combination therapy—27 patients (Dosage not specified) Duration undefined	N/A	Monotherapy: 25% Combination therapy: 26% (30-day)	Not addressed	Not addressed	Not addressed	In two-thirds of cases combo based on levofloxacin Non-comparative study
Ebara et al., 2017 [30]	South Korea—Retrospective cohort—Multicenter	From January 2007 to December 2013	Inclusion: Adults and pediatrics with SM BSI	BSI	Adults Male: 64%	Regimen: FQs—15 patients (Dosage not specified) Duration undefined 0% combination	Regimen: Minocycline—10 patients (Dosage not specified) Duration undefined 0% combination	N/A	FQs: 53% Minocycline: 40% (90-day)	Not addressed	Not addressed	Not addressed	Unpublished data of the original article were retrieved from the paper of Ko et al. [10] Non-comparative study
Kim SH et al., 2018 [31]	South Korea—Retrospective cohort—Single center	From January 2006 to December 2016	Inclusion: Adults, cancer patients; Exclusion: combination therapy	BSI	Adults Male: 59% Mean age: 55.7 years	Regimen: TMP/SMX—31 patients (Dosage not specified) Duration undefined 0% combination	Regimen: Levofloxacin—40 patients (Dosage not specified) Duration undefined 0% combination	N/A	TMP/SMX: 43% Levofloxacin: 36% (30-day)	Not addressed	Not addressed	Not addressed	Case-control study (controls being not-SM BSI) Non-comparative study
Velázquez-Acosta et al., 2018 [32]	Mexico—Retrospective cohort—Single center	From January 2000 to December 2016	Adult patients with BSI or pneumonia by SM	BSI: 55% Pneumonia: 45	Adults Male: 42% Mean age: 46.9 years Solid tumors: 63% Hematologic malignancies: 37%	Regimen: TMP/SMX—87 patients (Dosage not specified) Duration undefined 22% combination	Regimen: FQs—39 patients (Dosage not specified) Duration undefined Combination allowed	Regimen: Other—84 patients Not reported	TMP/SMX: 44% FQs: 18% Other: 24% (30-day)	Not addressed	Not addressed	No TMP/SMX use versus its use: aOR 0.87 (95% CI 0.3–2.65) Adjusting was made for age and appropriateness of therapy	Polymicrobial bacteremia: 20% (out of 95 BSI) All study population was composed of oncologic/ onco-hematologic patients Non-comparative study
Watson et al., 2018 [33]	United States—Retrospective cohort—Single center	From January 2004 to October 2014	Inclusion: patients at least 18 years of age that received at least 48 h of monotherapy with FQ or TMP/SMX. Exclusion: combination active therapy or therapy for less than 48 h.	BSI	Adults Male: 48% Mean age: 51.4 years (calculated combining two means) MV: 33%	Regimen: TMP/SMX—32 patients (Dosage not specified) Duration undefined 0% combination	Regimen: FQs—22 patients (Dosage not specified) Duration undefined 0% combination	N/A	TMP/SMX: 31% FQs: 14% (in-hospital)	TMP/SMX: 6% FQs: 5% (drug discontinuation)	TMP/SMX: 15 (IQR 7–38) days Levofloxacin:9 (IQR 5–16) days (hospital LOS)	Not addressed	Comparative study
Kim EJ, 2019 [34]	South Korea—Retrospective cohort—Multicenter	From January 2006 to December 2014	Inclusion: patients at least 18 years of age and positive blood culture for SM	BSI	Adults Solid tumor: 40% Hematological malignancy: 14	Regimen: TMP/SMX—31 patients (Dosage not specified) Duration undefined Combination allowed	Regimen: FQs—40 patients (Dosage not specified) Duration undefined Combination allowed	N/A	TMP/SMX: 87% FQs: 48% (60-day)	Not addressed	Not addressed	Not addressed	Non-comparative study
Nys et al., 2019 [35]	United States—Retrospective cohort—Single center	From January 2012 to October 2016	Inclusion: Adults Exclusion: polymicrobial infections.	Lung infection: 92% UTI: 3%.	Adults Male: 54% Median age: 63 (IQR 51–70) years MV: 37%	Regimen: TMP/SMX—45 patients (median dose 10.3 mg/kg/day) Median duration 13 days (IQR 8-15) 0% combination	Regimen: Levofloxacin—31 patients (median dose 750 mg/day) Median duration 13 days (8–15) 0% combination	N/A	TMP/SMX: 16% Levofloxacin: 13% (28-day)	TMP/SMX: 7% Levofloxacin: 0% (Adverse events)	Not addressed	Not addressed	Clinical cure (at the end of therapy): TMP/SMX = 82% Levofloxacin = 74% Comparative study
Shah et al., 2019 [36]	United States—Retrospective cohort	From November 2011 to October 2017	Patients with SM pneumonia. Exclusion: Less than 48 h of effective therapy.	Pneumonia	Adults Mean age 62 years (derived from combining group) Male: 62% Immunocompromised: 20% Polymicrobial pneumonia: 54%	Regimen: Monotherapy—214 patients TMP/SMX= 66% FQs = 30% Other = 4% (Dosage not specified) Duration undefined	Regimen: Combination therapy—38 patients TMP/STX + FQ = 50% TMP/STX + minocycline = 16% FQs + minocycline = 13% Duration undefined (Dosage not specified)	Not reported	Monotherapy: 23% Combination therapy: 40%% (30-day)	Not addressed	Monotherapy: 22 (IQR 14–35) days Combination therapy: 22.5 (IQR 14–44) days (hospital LOS)	Not addressed	Recurrence (30-day): Monotherapy = 8% Combination therapy = 11% Clinical cure (Improvement in signs and symptoms of infection after 7 days of effective therapy): Monotherapy = 60% Combination therapy = 53% Comparative study
Tokatly Latzer et al., 2019 [37]	Israel—Retrospective cohort—Multicenter	From 2012 to 2017	Patients hospitalized in pediatric ICU affected by BSI related to SM with or without a culture from a commonly sterile respiratory site	BSI: 42% CVC-related BSI: 22% BSI + Pleural fluid: 22%	Children younger than 18 years old. Oncologic: 22% Cerebral palsy: 22% Congenital cardiac disease: 15% Immunodeficiency: 9% End-stage renal disease: 7% Burss: 4%	Regimen: TMP/SMX—22 patients (Dosage not specified) Duration undefined Combination allowed	Regimen: Ciprofloxacin—13 patients (Dosage not specified) Duration undefined Combination allowed	Regimens Ciprofloxacin + TMP/SMX Ciprofloxacin + TMP/SMX + Minocycline Ceftazidime (Dosage not specified) Duration undefined Combination allowed	TMP/SMX: 27% Ciprofloxacin: 21% Ciprofloxacin + TMP/SMX: 10% Ciprofloxacin + TMP/SMX + Minocycline: 17% Ceftazidime: 14% (7-day)	Not addressed	Not addressed	Not addressed	Polymicrobial infection 37 (55%) When considering only monotherapy, just 35 cases were taken into account Non-comparative study
Alsuhaibani et al., 2021 [38]	Saudi Arabia,—Retrospective cohort—Single center	From January 2007 to December 2018	Inclusion: Pediatrics patients; Exclusion: asymptomatic patients, no therapy	BSI	Pediatrics. Male: 50% Under 12 months: 38% Malignancy: 29% Polymicrobial infection 30.9%	Regimen: TMP/SMX—36 patients (Dosage not specified) Duration undefined 0% combination	Regimen: TMP/SMX + others—11 patients (Dosage not specified) Duration undefined 100% combination	N/A	TMP/SMX: 31% TMP/SMX + others: 36% (7-day)	Not addressed	Not addressed	Not addressed	Comparative study (monotherapy versus combination therapy)
Junco et al., 2021 [39]	United States—Retrospective cohort—Multicenter	From January 2010 to January 2016	Inclusion: Adults; Exclusion: combination therapy, less than 48 h of monotherapy, patients with diagnosis of cystic fibrosis, resistance to initial therapy; SM infection in the previous 12 months	Pneumonia: 68%; BSI: 10%; UTI: 9%; ABSSSI: 11%; Other infections: 2%.	Adults Male: 61% Mean age: 59.6 years MV: 56%	Regimen: TMP/SMX—217 patients (median dose 9.7 mg/kg/day) Median duration 12 days 0% combination	Regimen: FQs—28 patients (Ciprofloxacin 800 mg/day or levofloxacin 750 mg/daily or moxifloxacin 400 mg/day) Median duration 12 days 0% combination	Regimen: Minocycline—39 patients (200 mg/day) Median duration 12 days 0% combination	TMP/SMX: 15% FQs: 29% Minocycline: 5% (30-day)	TMP/SMX: 47% FQs: 75% Minocycline: 74% (KDIGO AKI stage 1-2-3)	Median values TMP/SMX: 12 days (IQR 8–17) FQs: 12.5 days (IQR 8–19) Minocycline: 14 days (IQR 11–18) (infection-related LOS)	FQ use: aOR 0.3 (95% CI 0.1–2.1)— Adjusted for vasopressor support, APACHE, age, LOS prior to culture—FQ versus TMP/SMX Minocycline use: aOR 0.2 (95% CI 0.1–0–7)— Adjusted for vasopressor support, APACHE, age, LOS prior to culture-minocycline versus TMP/SMX)	Polymicrobial infection included but not specificied Clinical failure (isolation of SM from a subsequently collected culture from the same site of index culture after at least 48 h of therapy or alteration of monotherapy after at least 48 h of treatment for either an adverse event or concern for clinical failure or 30-day in-hospital all-cause mortality): TMP/SMX = 35% FQs = 29% Minocycline = 39% Comparative study (for the meta-analysis the “others” group comprised FQs plus TDs)
Puech et al., 2021 [40]	Reunion Island (French overseas department)—Retrospective cohort—Single center	From January 2010 to December 2018	Patients ICU-admitted with VAP by SM	100% VAP	Adults Male: 64% Median age: 61 [IQR 51–70] years Median SOFA: 9 [IQR 7–12] Immunoompromised: 5%; BSI: 3% Polymicrobial 58%	Regimen: TMP/SMX—80 patients (1200 mg/240 mg each 6 h) Duration undefined Combination allowed	Regimen: FQs—84 patients (ciprofloxacin 400 mg/8 h or moxifloxacin 400 mg/day) Duration undefined Combination allowed	Regimen (Other)—132 patients: Ticarcillin/ clavulanate 4 g/8 h; or ceftazidime 2 g/6 h Duration undefined Combination allowed	TMP/SMX: 50% FQs: 52% Ticarcillin/ clavulanate: 79% Ceftazidime 56% (in-hospital)	Not addressed	Not addressed	Not addressed	Monomicrobial infections in 55% cases. Monotherapy only in 4 patients (0.03%) Median MV duration: 21 [IQR 14–37] days Non-comparative study
Tuncel et al., 2021 [41]	Turkey—Retrospective cohort—Single center	From January 2002 to December 2016	Adult patients with nosocomial SM BSI	Catheter-related BSI: 21% Pneumonia: 7% Intraabdominal Infection: 6% Undetected source: 67%	Median (IQR) age: 54 (18–84) years Male: 58% ICU: 51%; Inpatient clinic: 49% Solid organ malignancy 30%.; Hematological malignancy 23%; Cerebrovascular disease: 17%; Multiple underlying diseases: 31%	Regimen: TMP/SMX—49 patients Duration undefined (Dosage not specified) Combination allowed	Regimen: Levofloxacin—17 patients Duration undefined (Dosage not specified) Combination allowed	Regimen: Other—28 patients	14-day mortality TMP/SMX: 22% Levofloxacin: 24% Other: 36% 30-day mortality TMP/SMX: 37% Levofloxacin: 24% Other: 55%	Not addressed	Not addressed	Not addressed	Polymicrobial infections: 34% Exclusion of 38 patients under TMP/SMX plus levofloxacin Non-comparative study
Zha et al., 2021 [42]	China—Retrospective cohort—Multicenter	From January 2017 to December 2020	Adult patients ICU-admitted with VAP by SM	100% VAP	Median (IQR) age = 76 (64.25–85) years Male: 79% Median APACHE II Score: 21 (IQR 16.25–24) Median Charlson index comorbidity score: 5 (IQR 4–6) Malignancy: 10 (12.2%)	Regimen: FQs—36 patients (dosage Levofloxacin 750 mg/daily; Moxifloxacin 400 mg/daily) 0% combination	Regimen: Tigecycline—46 patients (dosage: 100 mg followed by 50 mg × 2/daily) 0% combination	N/A	FQs: 28% Tigecycline: 48% (28-day)	Not addressed	Not addressed	Tigecycline versus FQs: aOR 1.64 (95% CI 0.58–4.77) Adjusting was made for the following variable: age, gender, chronic kidney disease, coagulation disorder, malignancy, polymicrobial infection, definitive antibiotic therapy, combination therapy with carbapenems, APACHE II score and Charlson comorbidity index score	Polymicrobial infections: 71% *A. baumannii*: 45% *P. aeruginosa*: 17% Clinical cure (complete resolution of all signs and symptoms of pneumonia at 14 days after the initial given dose of target antibiotics): FQs = 64% Tigecycline = 33% Comparative study
Ahlstrom et al., 2022 [43]	Denmark—Retrospective cohort—Single center	From January 2015 to June 2020	Patients with positive blood culture with detectable SM	100% BSI	Mainly adult patients with median age 41 (IQR 16–67) Male: 64% ICU: 23%	Regimen: TMP/SMX—48 patients (Dosage not specified) Duration undefined Combination allowed	Regimen: Ciprofloxacin —22 patients Duration undefined Combination allowed	N/A	TMP/SMX: 19% FQs: 18% (90-day)	Not addressed	Not addressed	TMP/SMX use: Adjusted HR 0.76 (95% CI 0.23–2.54)	14/48 of TMP/SMX patients received ciprofloxacin, 14/22 viceversa Non-comparative study
Sarzynski et al., 2022 [44]	United States—Retrospective cohort—Multicenter	From January 2005 to December 2017	Adult patients with BSI or LRTI by SM infection Exclusion: Inconsistent/no therapy	TMP/SMX: BSI = 8,4%; LRTI = 91.6% FQs: BSI = 12%; LRTI = 88%	Adults Male: 57% TMP-SMX median age: 60 [IQR, 31–72] years MV: 38.7% ICU stay: 33.5% Immunocompromised: 1% Levofloxacin: age 66 [IQR, 53–76] years MV: 31.2% ICU stay: 28.8% Immunocompromised: 1.7%	Regimen: TMP/SMX—758 patients (Dosage not specified) Duration undefined 0% combination	Regimen: Levofloxacin—823 patients (Dosage not specified) Duration undefined 0% combination	N/A	In-hospital: TMP/SMX =14.2% Levofloxacin = 10.6% Total mortality: TMP/SMX =17.7% Levofloxacin = 15.2%	Not addressed	TMP/SMX: 17 (9–31.8) days Levofloxacin:10 (5–21) days (hospital LOS)	FQs versus TMP/SMX: aOR 0.76 (95% CI 0.58–1.00). Adjusted values were computed using logistic regression after controlling for baseline patient and hospital level factors.	Polymicrobial infection: Levofloxacin = 42%, TMP/SMX = 42% Comparative study

Abbreviations: ABSSSI: Acute Bacterial Skin and Skin Structure Infection; AKI: cute kidney injury; aOR: adjusted odds ratio; BSI: bloodstream infection; CI: confidence interval; FQs = fluoroquinolones; HR: hazard ratio; ICU: intensive care unit; IQR: interquartile range; KDIGO: Kidney Disease: Improving Global Outcomes; IAI: intra-abdominal infection; LOS: length of stay; LRTI: lower respiratory tract infection; MV = mechanical ventilation; N/A: not applicable; OR: odds ratio; SM: Stenotrophomonas maltophilia; SSTI: skin and soft tissue infection; TMP/SMX = trimethoprim/sulfamethoxazole; UTI; urinary tract infection; VAP: ventilator-associated pneumonia.

**Table 2 antibiotics-12-00910-t002:** Outcomes for each comparison.

**Outcome: Mortality (All-Cause)**							
**Comparison**	**Included Studies**	**Number of Patients**	**OR, 95% CI**	**I²**	**Prediction Interval**	**E-Value**	**Comments**
TMP/SMX versus FQs	11	2407	1.46 (1.15–1.86)	33%	1.10–1.93	For point estimate: 1.71; for CI: 1.35.	See forest plot (Figure 2) for subgroup analysis about different timing of mortality. All monotherapy studies. One pediatric study [27]. FQs: five studies about levofloxacin [24,26,31,35,44], one about ciprofloxacin [37], five mixed [25,28,32,33,39].
TMP/SMX versus FQs-BSI	4	234	2.61 0.75–9.02	67%	0.01–503.12	For point estimate: 2.61; for CI: 1.	Different timing of mortality: 30-day [24], in-hospital [28,33], 7-day [37]. One pediatric study [37]. FQs: one study about levofloxacin [24], one about ciprofloxacin [37], two mixed [28,33].
TMP/SMX versus FQs not only monotherapy	15	2806	1.58 (1.10–2.27)	43%	0.58–4.35	For point estimate: 1.83; for CI: 1.28.	See forest plot for subgroup analysis about different timing of mortality (Figure 4). One pediatric study [37]. FQs: six studies about levofloxacin [24,26,31,35,36,44], two about ciprofloxacin [37,43], seven mixed [25,28,32,33,34,39,40].
TMP/SMX versus FQs not only monotherapy-BSI	7	469	2.45 (1.13–5.31)	59%	0.24–24.76	For point estimate: 2.51; for CI: 1.32.	Different timing of mortality: 30-day [24,41], in-hospital [28,33], 60-day [34], 90-day [43], 7-day [37]. One pediatric study [37]. FQs: two studies about levofloxacin [24,41], two about ciprofloxacin [37,43], three mixed [28,33,34]
TMP/SMX versus TDs	3	346	1.95 (0.79–4.82)	0%	0.01–685.99	For point estimate: 2.14; for CI: 1.	All monotherapy studies. 30-day mortality. TDs: minocycline in two studies [27,39]. tigecycline in the other [23].
TMP/SMX versus others	5	791	1.33 (0.74–2.37)	58%	0.22–8.14	For point estimate: 1.57; for CI: 1.	Different timing of mortality: 14-day [21], 30-day [32,39,41], in-hospital [40].
FQs vs TDs	3	174	0.80 (0.28–2.23)	28%	0.00–13,453.68	For point estimate: 1.48; for CI: 1.	Different timing of mortality: 28-day [42], 30-day [39], 90-day [30]. TDs: minocycline as monotherapy in two studies [30,39], tigecycline in the other one mostly in combination for VAP [42].
Monotherapy versus combination	4	438	0.71 (0.41–1.22)	0%	0.16–3.08	For point estimate: 1.66; for CI: 1.	See forest plot (Figure 5) for a subgroup analysis about different timing of mortality and population.
**Outcome: Mortality—Adjusted Effect Size**							
**Comparison**	**Included Studies**	**Number of Patients**	**OR, 95% CI**	**I²**	**Prediction Interval**	**E-Value**	**Comments**
FQs versus TMP/SMX	3	1912	0.73 (0.56–0.95)	0%	0.13–4.10	For point estimate: 1.62; for CI: 1.19.	All monotherapy studies (Figure 3).
**Outcome: Clinical Failure**							
**Comparison**	**Included Studies**	**Number of Patients**	**OR, 95% CI**	**I²**	**Prediction Interval**	**E-Value**	**Comments**
TMP/SMX versus FQs	3	360	0.94 (0.53–1.67)	0%	0.02–39.64	For point estimate: 1.21; for CI: 1.	All monotherapy studies. Different definitions of clinical failure.
TMP/SMX versus TDs	3	346	0.78 (0.24–2.54)	70%	0.00–659,171.29	For point estimate: 1.52; for CI: 1.	All monotherapy studies. Different definitions of clinical failure. TDs: minocycline in two studies [27,39], tigecycline in the other [23].
TMP/SMX versus Others	2	385	1.35 (0.77–2.35)	0%	Incalculable	For point estimate: 1.6; for CI: 1.	TMP/SMX always in monotherapy, comparator group based prevalently (89%) on various combination regimens. Different definitions of clinical failure.
FQs vs TDs	2	149	0.48 (0.15–1.54)	64%	Incalculable	For point estimate: 2.24: for CI: 1.	TDs: minocycline as monotherapy in one study [39], tigecycline in the other one mostly in combination for VAP [42]. Different definitions of clinical failure.
**Outcome: Safety-Adverse Events Onset**							
**Comparison**	**Included Studies**	**Number of Patients**	**OR, 95% CI**	**I²**	**Prediction Interval**	**E-Value**	**Comments**
TMP/SMX versus FQs	4	461	1.89 (0.26–13.60)	81%	0.00–7492.40	For point estimate: 2.09; for CI: 1.	All monotherapy studies. Definitions: “any adverse event” for 2 studies [24,35], drug discontinuation in another [33], acute kidney injury in the last one [39].
**Outcome: Length of Stay**							
**Comparison**	**Included Studies**	**Number of Patients**	**MD, 95% CI**	**I²**	**Prediction Interval**	**E-Value**	**Comments**
TMP/SMX versus FQs	5	2064	2.90 (−4.19–9.99)	84%	−14.25– 20.05	For point estimate: 1.56; for CI: 1.	All monotherapy studies except a minority of patients in Cho et al. [24] Infection-related LOS in Junco et al. [39]
TMP/SMX versus TDs (minocycline)	2	301	16.33 (−252.49– 285.15)	85%	Incalculable	For point estimate: 1.66; for CI: 1.	All monotherapy studies. Infection-related LOS in Junco et al. [39]

Abbreviations: CI: confidence interval; FQs: fluoroquinolones; LOS: length of stay; MD: mean difference; OR: odds ratio; TDs: tetracycline derivatives; TMP/SMX: trimethoprim/sulfamethoxazole; VAP: ventilator-associated pneumonia.

## Data Availability

The datasets informing the current meta-analysis are available from the corresponding author upon reasonable request. They are derived from already published studies. All data analyzed for the meta-analysis are included in the corresponding published articles as reported in Table 1.

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
