# Peer review of "Stenotrophomonas maltophilia Infections: A Systematic Review and Meta-Analysis of Comparative Efficacy of Available Treatments, with Critical Assessment of Novel Therapeutic Options"

_antibiotics, 2023, doi:10.3390/antibiotics12050910_

Round 1

Reviewer 1 Report

Reviewer comment

Manuscript ID: antibiotics-2372410

 General concept comments

Stenotrophomonas maltophilia infection incidence is rare. Therefore only reported in case-control or cohort studies in health facilities supported with good laboratory microbiology. This type of bacteria will be considered in untreated infection.

 Specific comments

 Discussion and conclusion

RCT study is not an option because it is a rare infection and not ethical. In conclusion, the author could report any (patient) factor that lead to Stenotrophomonas maltophilia infections.

Minor editing of English language required

Author Response

We thank the referee for her/his general favourable comments.

Minor language edits were made also according to other referees' comments.

Risk factor for SM infections have been already described in the text (please see introduction).

The necessity of RCT is stated. The question is not ethical but the difficult process to recruit patients affected by this not frequent condition.

Reviewer 2 Report

Alberto et al did a meta-analysis and systematic review to evaluate the treatments of Stenotrophomonas maltophilia infections. The study suggested that TDs could be a reasonable alternative to TMP/SMX for the treatment of SM infections. This manuscript was well-written and prepared. Detailed analysis and comparison were performed. I would suggest a minor revision.

Please address the below concerns and correct the minor errors:

1.      Line 15, is it possible to get the most updated search? It has been one year till now.

2.      Line 19 and 22, 244, 294, I2, “2” should be superscript, and “I” is italic.

3.      Line 155, is duplicates number 716 right?? After deduction that, is 1572 records, and plus 4 handing-searches, is 1576.

4.      Line 156, should be citation [10].

5.      Line 160, All Figure legends should be at the bottom in general, please follow the instruction of the journal. Same in the Supplementary data.

6.      Line 169, I assume should be the study that contributed... not “the”.

7.      Table 2, column width didn’t fit well in the last raw.

8.      There is line space problem in some paragraphs, please be consistent.

9.      Table S4a, please align those star symbols.

10.  Line 323. Should be Citation [28].

11.  Both part 4 and 5 are Discussion???

Author Response

We thank the referee for her/his general favourable comments.

A point-by-point reply follows (in red and italics):

  1. Line 15, is it possible to get the most updated search? It has been one year till now.  We thank the reviewer for this consideration. According to Cochrane (https://training.cochrane.org/handbook/current/chapter-04), we updated the search up to November 2022 (additional 8 months to be no older than 6 months).
  2. Line 19 and 22, 244, 294, I2, “2” should be superscript, and “I” is italic. We used the superscript whenever needed. The italics is not mandatory.
  3. Line 155, is duplicates number 716 right?? After deduction that, is 1572 records, and plus 4 handing-searches, is 1576. We rerun the search so the figures have changed.
  4. Line 156, should be citation [10]. We have amended it.
  5. Line 160, All Figure legends should be at the bottom in general, please follow the instruction of the journal. Same in the Supplementary data. We will modify it with the help of the editorial team if the manuscript is accepted.
  6. Line 169, I assume should be the study that contributed... not “the”.We have amended it.
  7. Table 2, column width didn’t fit well in the last raw. We will modify it with the help of the editorial team if the manuscript is accepted..
  8. There is line space problem in some paragraphs, please be consistent. We have amended it whenever possible.
  9. Table S4a, please align those star symbols. We have amended it whenever possible.
  10. Line 323. Should be Citation [28]. We have amended it. 

  1. Both part 4 and 5 are Discussion??? We have amended it. The paragraph is called "new therapeutic options".

Reviewer 3 Report

Excellent manuscript, both in the choice of objective and the quality of the conduct of the research.

No elements to be edited in a different way or to be studied further are identified and its publication is recommended as it is in total agreement with the objectives of the journal

Author Response

We thank the reviewer for his/her general very favourable comments.